# Extreme adaptations for aquatic ectoparasitism in a Jurassic fly larva

**Jun Chen[1†], Bo Wang[2,3]\*[†], Michael S Engel[4†], Torsten Wappler[2], Edmund A Jarzembowski[3,5], Haichun Zhang[3], Xiaoli Wang[1], Xiaoting Zheng[1], Jes Rust[2]**

[1]Institute of Geology and Paleontology, Linyi University, Linyi, China; [2]Steinmann Institute, University of Bonn, Bonn, Germany; [3]State Key Laboratory of Palaeobiology and Stratigraphy, Nanjing Institute of Geology and Palaeontology, Chinese Academy of Sciences, Nanjing, China; [4]Department of Ecology and Evolutionary Biology, University of Kansas, Lawrence, United States; [5]Department of Earth Sciences, Natural History Museum, London, United Kingdom

**Abstract** The reconstruction of ancient insect ectoparasitism is challenging, mostly because of the extreme scarcity of fossils with obvious ectoparasitic features such as sucking-piercing mouthparts and specialized attachment organs. Here we describe a bizarre fly larva (Diptera), *Qiyia jurassica* gen. et sp. nov., from the Jurassic of China, that represents a stem group of the tabanomorph family Athericidae. *Q. jurassica* exhibits adaptations to an aquatic habitat. More importantly, it preserves an unusual combination of features including a thoracic sucker with six radial ridges, unique in insects, piercing-sucking mouthparts for fluid feeding, and crocheted ventral prolegs with upward directed bristles for anchoring and movement while submerged. We demonstrate that *Q. jurassica* was an aquatic ectoparasitic insect, probably feeding on the blood of salamanders. The finding reveals an extreme morphological specialization of fly larvae, and broadens our understanding of the diversity of ectoparasitism in Mesozoic insects.

**\*For correspondence:**
savantwang@gmail.com

[†]These authors contributed equally to this work

**Competing interests:** The authors declare that no competing interests exist.

## Introduction

The early evolution of insect ectoparasites and their associations with hosts are poorly known (*Labandeira, 2002*; *Wappler et al., 2004*; *Grimaldi and Engel, 2005*). Although several Mesozoic insects were regarded as putative ectoparasites, only giant fleas have been widely accepted as definite terrestrial ectoparasitic insects on dinosaurs, pterosaurs, or mammals (*Gao et al., 2012*, *2013b*; *Huang et al., 2012*, *2013*). Here we report on an aquatic ectoparasitic insect based on five well-preserved specimens from the Middle Jurassic Daohugou beds of China. These fossils are extremely rare among the approximately 300,000 fossil insects in the collections of the Nanjing Institute of Geology and Palaeontology and Shandong Tianyu Museum of Nature.

## Results

### Systematic paleontology

Order Diptera Linnaeus, 1758
 Family Athericidae Stuckenberg, 1973
 *Qiyia jurassica* gen. et sp. nov.

### Etymology

*Qiyia* is from the Chinese 'qiyi' meaning bizarre; *jurassica* is a reference to the Jurassic age of the fossils.

**eLife digest** Parasites have been exploiting other organisms for millions of years. However, little is known about ancient parasitic insects, as it is rare to find fossils that are preserved well enough for them to be identified as parasites. This is particularly true for ectoparasitic insects, which live on the skin of their hosts. As a result, the only widely accepted ectoparasitic insect from the Mesozoic era is the giant flea, which infested dinosaurs, pterosaurs or mammals.

Now, Chen, Wang, Engel et al. have discovered a new genus and species of ancient aquatic fly, which may be the earliest currently known aquatic ectoparasitic insect. Named *Qiyia jurassica*—after the Chinese word for 'bizarre' and the Jurassic period when it lived—its larva has a combination of features that mark it out as a parasitic ancestor of modern water snipe flies. In addition, the well-preserved fossilised larvae used to identify *Q. jurassica* have some more unusual features.

The mouth of *Q. jurassica* had a structure commonly found in ectoparasites, designed to pierce skin and suck blood. The larva also had several features that were particularly well-adapted for gripping the host animal while underwater. The prolegs—stumpy fleshy structures found on the abdomen—were covered in bristles that pointed upwards, anchoring the larva in place. *Q. jurassica* also had an unusual sucker on its thorax that would have provided a firm grip that held its head still during feeding. Although many modern aquatic ectoparasites—like leeches—have suckers, the *Q. jurassica* sucker may be unique amongst insect larvae, as it has six large ridges and is covered in spines. Both features may have provided extra grip.

Chen, Wang, Engel et al. suggest that *Q. jurassica* feasted on the blood of salamanders, as many salamander fossils have been found in the same region. The larvae could have attached to unexposed areas of the salamander—behind the gills, for example—where feeding would also have been easier due to the rich supply of blood vessels, and the thinner, more easily pierced skin.

The wide range of features found on *Q. jurassica* suggests that Mesozoic ectoparasitic insects were more diverse than previously thought.

## Type material

Holotype STMN65-1. Paratypes STMN65-2, NIGP156982, NIGP156983, NIGP156984. All specimens are preserved as carbonaceous impressions on the surface of grey tuffaceous siltstone (*Wang et al., 2013*).

## Locality and age

From the Middle Jurassic Daohugou beds (approximately 165 million years old) of Ningcheng County, Inner Mongolia, China (*Liu et al., 2006*).

## Diagnosis

Three thoracic segments fused, with a ventral sucker; two pairs of dorsal spines on abdominal segments 1–7; abdominal segments 1–6 with paired ventral prolegs bearing upward directed bristles and apical crochets; extended seventh proleg; two pairs of anal papillae; sclerotized terminal processes with stiff setae.

## Description

Body elongate, 18–24 mm long (*Table 1*). Head greatly reduced and partly retractile into thorax (*Figure 1A,B*); antennae and eyes not visible (*Figure 1C*); a pair of sclerotized tentorial rods (*Figure 2B*). Mandibles approximately 0.6 mm long, heavily sclerotized, sickle-shaped, oriented to move parallel to each other in vertical plane, with external groove on adoral surface extending whole length of mandible (*Figure 1E*). Thoracic segment swollen, slightly narrower than abdomen (*Figure 2A*). Sucker retractile, diameter about 2 mm, located ventrally on thoracic segment and consisting of a circular suction disc with central opening about one quarter of disc diameter; peripheral area of disc thin and flexible (*Figure 1D*). Six robust, sclerotized ridges on sucker, radially arranged, covered by soft skin with small spines (*Figure 2D,E*); distal part of each ridge thickened, probably with three processes embedded in musculature (*Figure 2E*). Three pairs of small spines with simple shafts on dorsolateral margins of thorax, two pairs on dorsolateral margins of abdominal segments 1–7, and one pair

**Table 1.** Measurements of specimens of *Qiyia jurassica*

| | Holotype STMN65-1 | Paratype STMN65-2 | Paratype NIGP156982 | Paratype NIGP156983 | Paratype NIGP156984 |
|---|---|---|---|---|---|
| Orientation | lateral | lateral | dorsal | dorsal | lateral |
| Body | 23.8 | 22.1 | 22.9 | ~22 | 18.1 |
| Head | ~1 | ~1 | ~1 | – | 0.8 |
| Thorax | 2.8 | 2.5 | 2.6 | ~2.5 | 2.3 |
| Thoracic sucker diameter | 2.0 | 1.9 | – | – | 1.6 |
| Ridge | 0.6 | 0.6 | – | – | 0.5 |
| Abdominal segments 1–7 (average) | ~2.3 | ~2.2 | ~2.3 | ~2.2 | ~1.9 |
| Prolegs 1–6 (average) | ~1.5 | ~1.5 | ~1.5 | ~1.5 | ~1.3 |
| Seventh proleg | 4.0 | 3.8 | – | – | 3.0 |
| First anal papilla | 6.1 | 6.0 | – | ~6 | 4.8 |
| Second anal papilla | 3.7 | 3.2 | – | – | – |
| Terminal process | 2.9 | 2.7 | 3.0 | 2.7 | 2.3 |

All measurements are in mm and lengths except where otherwise indicated.
~: approximately; –: unknown.

on abdominal segment 8 (*Figure 2A*). Abdomen with eight distinct segments, covered by many short setae. Abdominal segments 1–6 with a pair of cylindrical, ventral prolegs covered by stiff, upward directed bristles; each proleg nearly half width of body with two rows of six crochet hooks apically (*Figure 1F*); seventh proleg approximately three times longer than other prolegs with only three or four apical hooks; abdominal segment 8 with two pairs of slender, tapering anal papillae: first pair long, approximately quarter body length; second pair half the length of the first pair (*Figure 1A,B*); one pair of unsegmented, sclerotized terminal processes fringed with stiff setae, approximately one-tenth body length; each process with about 10 spiracles (*Figure 1G*, *Figure 2C*).

## Discussion

Three specimens are laterally compressed (STMN65-1, STMN65-2, NIGP156984) and two are dorso-ventrally compressed (NIGP156982, NIGP156983), thereby providing side and top views of the detailed morphology of the larva. *Q. jurassica* is attributed to the Tabanomorpha by the reduced and retractable head and sickle-shaped mandibles shifted into a vertical plane (*Yeates, 2002*; *Zloty et al., 2005*). It possesses two noticeably plesiomorphic features: mandibles with external grooves (*Zloty et al., 2005*) and well-developed anal papillae (*Wichard et al., 1999*), while sharing two potential synapomorphies with extant athericid larvae: paired prolegs with crochet hooks (*Yeates, 2002*; *Kerr, 2010*) and long terminal processes fringed with setae (*Dobson, 2013*). This combination of primitive and derived features demonstrates that *Q. jurassica* is a stem lineage representative of the Athericidae (water snipe flies), a family sister to the more familiar horse flies (Tabanidae). The earliest known Athericidae and Tabanidae (all represented by preserved adults) are from the Early Cretaceous of southern England (*Mostovski et al., 2003*). Our new fossils are the earliest record of athericid flies and extend the lineage back to the Middle Jurassic, an age which is consistent with predicted divergence times based on molecular studies (estimated at the Early or Middle Jurassic) (*Wiegmann et al., 2011*).

*Q. jurassica* displays adaptations to an aquatic habitat, much like extant Athericidae which are today aquatic predators in fast-flowing water (as adults some athericids feed on mammalian or amphibian blood) (*Mostovski et al., 2003*; *Nagatomi and Stuckenberg, 2004*). The paired sclerotized terminal processes are morphologically comparable to the modifications of beetle urogomphi in the aquatic larvae of some families such as Dytiscidae (*Wichard et al., 1999*). About 10 spiracles are present on each process of *Q. jurassica* (*Figure 1G*, *Figure 2C*), confirming that they were used for breathing air, functionally similar to the unsclerotized ones of extant athericid larvae (*Nagatomi and Stuckenberg, 2004*). *Q. jurassica* also possesses two pairs of anal papillae which are useful for extracting dissolved oxygen from water in aquatic flies and also play an important part in salt absorption to maintain ionic

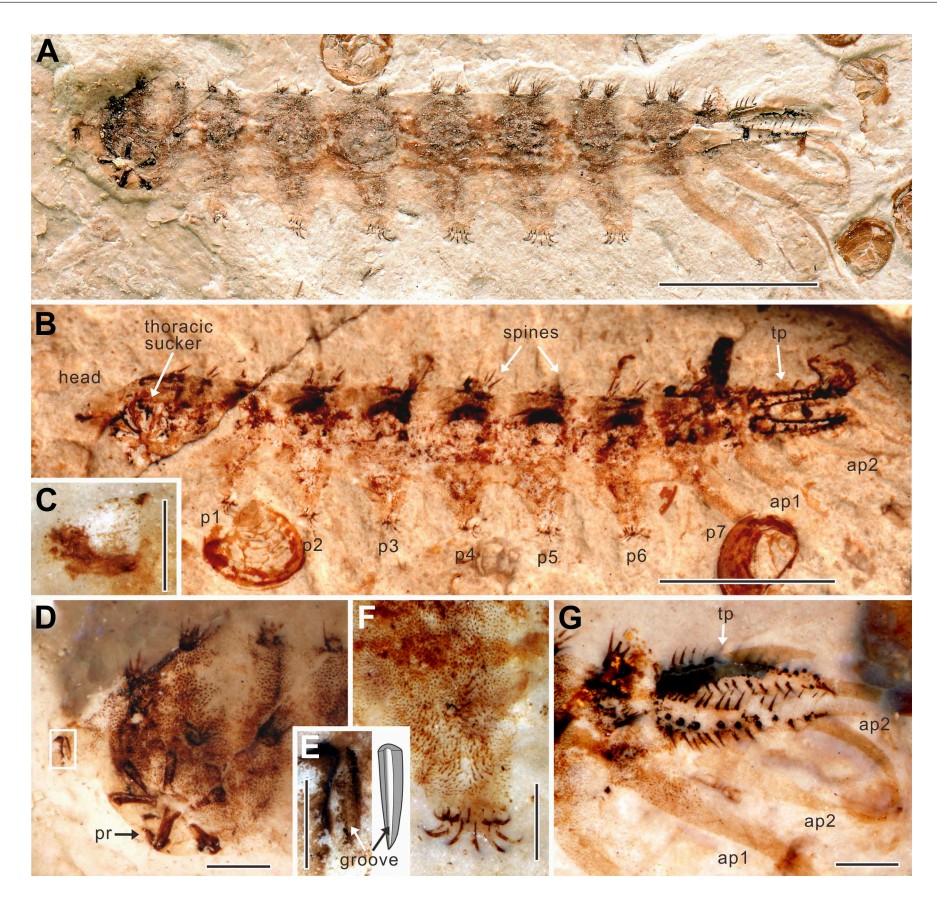

**Figure 1**. *Qiyia jurassica* from the Middle Jurassic epoch of Daohugou, China. (**A**) Holotype STMN65-1.
(**B**) Paratype STMN65-2 under alcohol (horizontal mirror image). (**C**) Head capsule of paratype STMN65-2. (**D**) Head
and thorax of holotype STMN65-1. (**E**) Enlargement and reconstruction of the mandible of holotype STMN65-1;
note the longitudinal groove. (**F**) Fifth proleg of holotype STMN65-1; note stiff, upward directed bristles which
are distinctly longer than setae on body. (**G**) Last abdominal segment of holotype STMN65-1. ap, anal papilla;
p, proleg; pr, process of ridge; tp, terminal process. (Scale bars: 5 mm in **A**, **B**, 1 mm in **D**, **F**, **G**, and 0.5 mm in **C**, **E**).

concentrations in the body fluids (*Wichard et al., 1999*). These organs are common in nematoceran
larvae and in some lower brachyceran larvae, but are reduced in extant tabanomorphan larvae
(*Wichard et al., 1999*; *Dobson, 2013*). In the case of the fossil larva, their development implies a ple-
siomorphic condition.

The most notable structure of these newly discovered fossils is the ridged thoracic sucker which is
a unique evolutionary adaptation among holometabolous insects. The round sucker has six radial
ridges which are considered to be highly modified thoracic legs (*Figure 2D*). These six robust, sclero-
tized ridges could increase both the suction area and surface friction, thus providing more adhesion
and increasing lateral stability whilst reducing slippage, like the radial grooves in modern octopus
suckers (*Kier and Smith, 2002*) and supporting ribs in man-made suction cups (*Monkman et al.,
2007*). The dense vestiture of small spines may be used for better anchoring on the corrugated skin of
a salamander, like the sucker-ring teeth and knobs on squid suckers (e.g., *Miserez et al., 2009*). To our
knowledge, among insect larvae, only extant blephaericids (Diptera) have six well-developed suckers,
but these are small and without ridges on the abdominal sternites. As blephaericid larvae graze on
periphyton on rocks, they use the suckers to adhere to the substrate in fast-flowing streams (*Frutiger,
2002*). However, the excellent preservation of our new fossils suggests that *Q. jurassica* did not travel
long distances and, unlike crown group Athericidae, most probably lived in still water near to or in the
Daohugou palaeolake, a low-energy preservation environment (*Wang et al., 2013*). The thoracic
sucker on *Q. jurassica* is strongly cephalad on the body so, when anchored to the substrate, it probably

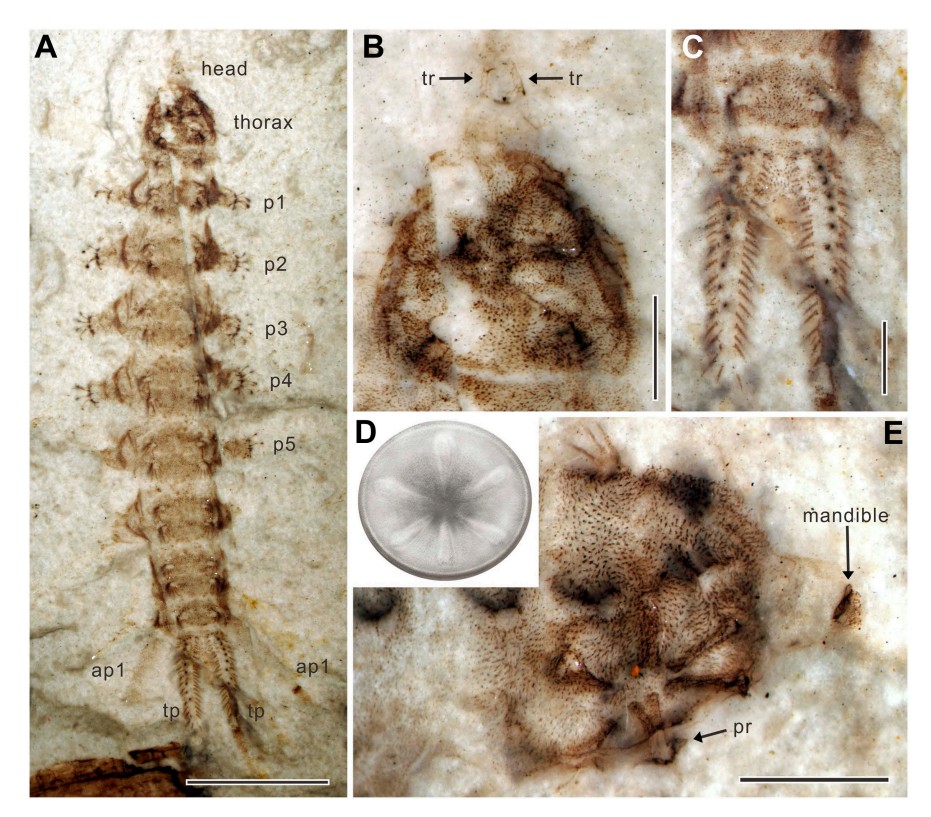

**Figure 2**. *Qiyia jurassica* from the Middle Jurassic epoch of Daohugou, China. (**A**) Paratype NIGP156982 under alcohol. (**B**) Head and thorax of paratype NIGP156982; note the underlying thoracic sucker. (**C**) Terminal processes of paratype NIGP156982. (**D**) Reconstruction of sucker. The sucker consists of a circular suction disc with central opening and thin peripheral area. Six robust, radially arranged ridges are covered by soft skin with small spines. (**E**) Head and thorax of paratype NIGP156984; note the deformed mandible. ap, anal papilla; p, proleg; pr, process of ridge; tp, terminal process; tr, tentorial rod. (Scale bars: 5 mm in **A**, 1 mm in **B**, **C**, **E**).

restricted the movement of the small, short head (*Figure 1D*, *Figure 2E*), a condition that is clearly suitable for piercing and sucking (*Figure 3*). Suckers are widespread in aquatic ectoparasites such as leeches, fish lice, and lampreys (*Kearn, 2004*) which require more suction power to avoid becoming dislodged; other aquatic ectoparasites without attachment organs embed themselves in skin or muscle, such as cyclopoid copepods (anchor worms) (*Kearn, 2004*). In addition to the sucker, the stiff, upward directed bristles and apical hooks on the prolegs (*Figure 1F*) are also specialized attachment structures. These morphological adaptations provide compelling evidence that *Q. jurassica* adhered to a host as an ectoparasite, providing further specialization for a dense, watery habitat.

Bloodsucking is considered to have evolved independently at least 12 times in true flies (*Lukashevich and Mostovski, 2003*; *Wiegmann et al., 2011*). It started with free-living scavengers or predators which subsequently became opportunistic feeders on vertebrates, such as the notorious Congo floor maggot (*Auchmeromyia*) that consumes the blood of sleeping humans (*Lehane, 2005*). Bloodsuckers are present as adults in three families of extant Tabanomorpha (*Nagatomi and Stuckenberg, 2004*). Although hitherto known larval Tabanomorpha are mainly predators, some larvae suck the body fluids of vertebrates such as anurans (*Jackman et al., 1983*). Predatory fly larvae commonly have morphological and physiological adaptations (such as efficient protein-digesting enzymes and salivary glands), facilitating the switch to bloodsucking (*Balashov, 1984*; *Lehane, 2005*). *Q. jurassica* has a pair of sickle-shaped mandibles with external grooves (*Figure 1E*), which is a ground-plan character of Tabanomorpha (*Wichard et al., 1999*; *Yeates, 2002*), forming a channel when the left and right mandibles are occluded (*Zloty et al., 2005*) and used for sucking blood or other body fluids (*Marshall, 1981*).

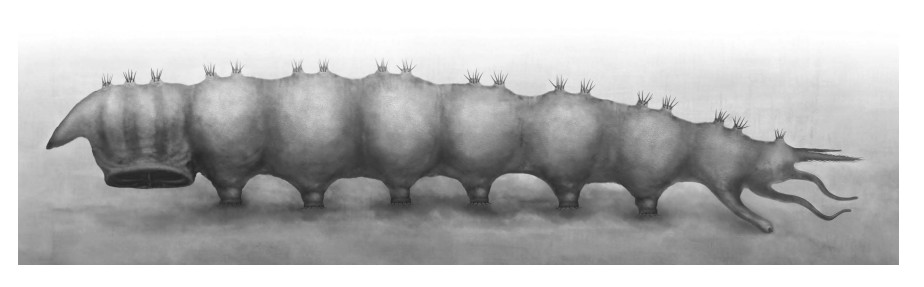

**Figure 3**. Reconstruction of *Qiyia jurassica* in lateral view.
The following figure supplements are available for figure 3:

**Figure supplement 1**. Ecological restoration of *Qiyia jurassica* from the Middle Jurassic epoch of Daohugou, China.

In the Daohugou deposits fish are completely absent but salamanders are extremely abundant (several thousand specimens recovered to date) (*Liu et al., 2006*). The most common salamanders at Daohugou, *Chunerpeton tianyiensis* and *Jeholotriton paradoxus*, have body lengths of 500 mm and 150 mm, respectively (*Wang and Rose, 2005*). Both species display neotenic features and are fully aquatic in all stages of their life cycle (*Gao et al., 2013a*). Salamander skin is glabrous and thin, and could easily have been penetrated by the mandibles of a larva such as *Q. jurassica*. The Daohugou salamanders match *Q. jurassica* well in size as well as co-occurrence, suggesting a possible parasite-host relationship. Some extant fly larvae parasitize anurans by burrowing into the skin, including Calliphoridae, Sarcophagidae, and Chloropidae (*Hoskin and McCallum, 2007*), and sometimes cause substantial mortality in their hosts (*Bolek and Coggins, 2002*). *Q. jurassica*, however, could simply have been anchored to the salamander skin using its sucker and prolegs (*Figure 3—figure supplement 1*), in a similar manner to leeches and fish lice (*Kearn, 2004*).

Despite a great taxonomic diversity of extant ectoparasitic insects (*Marshall, 1981*), previous definite Mesozoic records were confined to the terrestrial giant fleas from the Middle Jurassic and Early Cretaceous epochs (*Gao et al., 2012*, *2013b*; *Huang et al., 2012*). *Q. jurassica*, which is arguably the earliest known aquatic ectoparasitic insect, reveals an unexpected morphological specialization of fly larvae and highlights the diversity of ectoparasitism in the Mesozoic.

## Materials and methods

The specimens are housed in the Shandong Tianyu Museum of Nature (STMN), Pingyi, China, and Nanjing Institute of Geology and Palaeontology (NIGP), Chinese Academy of Sciences. Photographs were taken using a Zeiss Discovery V8 microscope system with specimens moistened in 95% alcohol or dry. The figures were prepared with CorelDraw X4 and Adobe Photoshop CS3.

### Nomenclatural acts

The electronic edition of this article conforms to the requirements of the amended International Code of Zoological Nomenclature, and hence the new names contained herein are available under that Code from the electronic edition of this article. This published work and the nomenclatural acts it contains have been registered in ZooBank, the online registration system for the ICZN. The ZooBank LSIDs (Life Science Identifiers) can be resolved and the associated information viewed through any standard web browser by appending the LSID to the prefix 'http://zoobank.org/'. The LSID for this publication is: urn:lsid:zoobank.org:pub: 99FE7164-CF29-4EAE-B7B2-40C727CAC4FA. The electronic edition of this work was published in a journal with an ISSN, and has been archived and is available from the following digital repositories: PubMed Central, CLOCKSS, Linyi University, Steinmann Institute at University of Bonn, and Nanjing Institute of Geology and Palaeontology (CAS). Printed copies are deposited in six major publicly accessible libraries including Linyi University, Nanjing Institute of Geology and Palaeontology (CAS), Steinmann Institute at University of Bonn, University of Kansas, Natural History Museum (London), and Muséum National d'Histoire Naturelle in Paris.

## Acknowledgements

We are grateful to Xing Xu, Yuan Wang, Alexandr Rasnitsyn, and Junfeng Zhang for discussions on these specimens, Daran Zheng for his help in the preparation of specimens, Dinghua Yang for reconstructions, and André Nel and Enrique Peñalver for their constructive reviews.

Additional information

This research was supported by the National Basic Research Program of China (2012CB821900), National Natural Science Foundation of China (41272013, 41372014, J1210006), Natural Scientific Foundation of Shandong Province (ZR2013DQ017), CAS 2011T2Z04, and partially from US National Science Foundation grant DEB-0542909. BW was also supported by a Research Fellowship from the Alexander von Humboldt Foundation.

## Additional information

### Funding

| Funder | Grant reference number | Author |
|---|---|---|
| Alexander von Humboldt Foundation | CHN 1149090 STP | Bo Wang |
| National Basic Research Program of China | 2012CB821900 | Jun Chen, Bo Wang, Edmund A Jarzembowski, Haichun Zhang, Xiaoli Wang, Xiaoting Zheng |
| National Natural Science Foundation of China (NSFC) | 41272013, 41372014, J1210006 | Jun Chen, Bo Wang, Edmund A Jarzembowski, Haichun Zhang, Xiaoli Wang, Xiaoting Zheng |
| Natural Scientific Foundation of Shandong Province | ZR2013DQ017 | Jun Chen |
| National Science Foundation (NSF) | DEB-0542909 | Michael S Engel |
| Chinese Academy of Sciences (CAS) | CAS 2011T2Z04 | Edmund A Jarzembowski |

The funders had no role in study design, data collection and interpretation, or the decision to submit the work for publication.

### Author contributions

JC, MSE, Acquisition of data, Analysis and interpretation of data, Drafting or revising the article, Contributed unpublished essential data or reagents; BW, Conception and design, Acquisition of data, Analysis and interpretation of data, Drafting or revising the article, Contributed unpublished essential data or reagents; TW, EAJ, Analysis and interpretation of data, Drafting or revising the article; HZ, XW, XZ, Acquisition of data, Analysis and interpretation of data; JR, Acquisition of data, Analysis and interpretation of data, Drafting or revising the article

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
