## [Decision Letter]

Thank you for sending your work entitled “Extreme adaptations for aquatic ectoparasitism in a Jurassic fly larva” for consideration at *eLife*. Your article has been favorably evaluated by Detlef Weigel (Senior editor) and 2 peer reviewers: André Nel and Enrique Peñalver.

The Senior editor and the reviewers discussed their comments before we reached this decision, and the Senior editor has assembled the following comments to help you prepare a revised submission:

This report of the earliest known aquatic ectoparasitic insect, from the Jurassic, is of great significance. The morphology and adaptations of these Jurassic larvae are fascinating. The specimens studied are impressive in their fine preservation, not leaving any doubt about their bizarre features. The paper is well written, the arguments are solid; the illustrations are nice and convincing. The detailed descriptions and their interpretations are completely convincing, despite all that seems very strange at first glance. It is a superb contribution to the knowledge of the paleobiology and evolution of the insects.

Minor comments:

1) Please indicate the family of these larvae. According to the main text “Our new fossils are the earliest record of athericid flies...” It appears that the adscription to this family is completely clear.

2) Maybe there is a more suitable word (currently teeth) to name the 6 strongly sclerotized structures on the ventral sucker...

3) The presence of a dense vestiture of small spines is in contradiction with a good functionality of a sucker due to difficulty to avoid the entrance of water (therefore, internal pressure loss), thus maybe for this reason the structure contains six “teeth” to improve the adherence.

4) Spiracles mentioned in the Discussion section are absent in the Description section.

5) The figure of the ecological restoration correctly reflects the authors' most plausible interpretation. One should to note that most probably these larvae could be located on salamander body zones that are not very exposed, since other salamanders could otherwise prey on them.

---

## [Author Response]

*1) Please indicate the family of these larvae. According to the main text “Our new fossils are the earliest record of athericid flies...” It appears that the adscription to this family is completely clear*.

Thanks. We have added it.

*2) Maybe there is a more suitable word (currently teeth) to name the 6 strongly sclerotized structures on the ventral sucker..*.

Thanks. Done. We have changed “tooth” and “teeth” to “ridge” and “ridges” respectively.

*3) The presence of a dense vestiture of small spines is in contradiction with a good functionality of a sucker due to difficulty to avoid the entrance of water (therefore, internal pressure loss), thus maybe for this reason the structure contains six “teeth” to improve the adherence*.

We think a vestiture of small spines is not in contradiction with a good functionality of a sucker. These small spines may be used for better anchoring on a corrugated skin of a salamander, like the sucker-ring teeth and knobs on squid suckers, thus increasing the functionality of the sucker. We have added the following to the Discussion: “The dense vestiture of small spines may be used for better anchoring on a corrugated skin of a salamander, like the sucker-ring teeth and knobs on squid suckers (e.g., Miserez et al., 2009).”

*4) Spiracles mentioned in the Discussion section are absent in the Description section*.

Thanks a lot. We have added the description of spiracles: “each process with about ten spiracles.”

*5) The figure of the ecological restoration correctly reflects the authors' most plausible interpretation. One should to note that most probably these larvae could be located on salamander body zones that are not very exposed, since other salamanders could otherwise prey on them*.

We fully agree with the reviewers that these larvae could be located on unexposed body zones, such as on the axil or behind the gill, where there are many blood vessels and the skin is thinner. In the restoration, we put the larva on the area near the gills to help the reader see the larva and show the association between the larva and salamander. To clarify this, we have added the following to the figure legend: “Larvae could be located on unexposed body zones, such as on the axil or behind the gill, where there are many blood vessels and the skin is thinner.”